# "Why Not Other Classes?": Towards Class-Contrastive Back-Propagation Explanations

**Yipei Wang, Xiaoqian Wang**[*]
Elmore Family School of Electrical & Computer Engineering
Purdue University
West Lafayette, IN 47906
{wang4865,joywang}@purdue.edu

## Abstract

Numerous methods have been developed to explain the inner mechanism of deep neural network (DNN) based classifiers. Existing explanation methods are often limited to explaining predictions of a pre-specified class, which answers the question "why is the input classified into this class?" However, such explanations with respect to a single class are inherently insufficient because they do not capture features with class-discriminative power. That is, features that are important for predicting one class may also be important for other classes. To capture features with true class-discriminative power, we should instead ask "why is the input classified into this class, *but not others*?" To answer this question, we propose a weighted contrastive framework for explaining DNNs. Our framework can easily convert any existing back-propagation explanation methods to build class-contrastive explanations. We theoretically validate our weighted contrast explanation in general back-propagation explanations, and show that our framework enables class-contrastive explanations with significant improvements in both qualitative and quantitative experiments. Based on the results, we point out an important blind spot in the current explainable artificial intelligence (XAI) study, where explanations towards the predicted logits and the probabilities are obfuscated. We suggest that these two aspects should be distinguished explicitly any time explanation methods are applied.

## 1 Introduction

The black-box essence of Deep Neural Networks (DNNs) has been impeding the development and application of powerful deep learning tools in reality. Without providing a promising reasoning process, DNNs are never fully trusted by end-users, especially for high-stake areas such as medical analysis, autonomous vehicles, etc. Recently, General Data Protection Regulation (GDPR) even explicitly proclaim the right to explanations for automated decisions (Selbst and Powles, 2018). This further stimulates the demand for explainable DNNs. As a result, the studies of eXplainable Artificial Intelligence (XAI) thrive to explore the inner mechanism of DNNs and to build trustworthy models. Since then, countless explanation methods have been devised to serve such purposes. Methods developed have covered various aspects including pattern recognition, decision making, natural language inference, etc. Albeit seemingly prosperous, the lack of systematicness is an inevitable issue of XAI studies. One of the fundamental questions to be answered is *what makes a proper explanation to the black-box model?* We need to know what end users are longing for before we develop the techniques. For the moment, this still remains an open question as there is no unified framework for XAI. A reasonable workaround is to set specific goals (Wang and Wang, 2022) and

---

[*]Corresponding author.

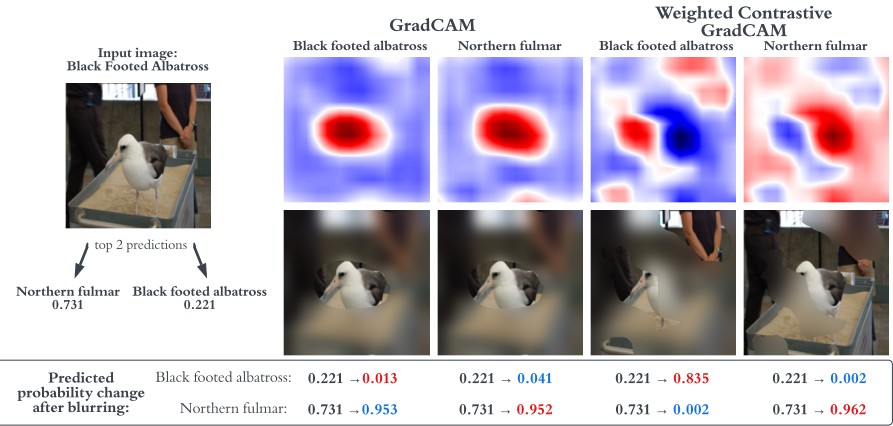

| | | GradCAM | | Weighted Contrastive GradCAM | |
|---|---|---|---|---|---|
| | | Black footed albatross | Northern fulmar | Black footed albatross | Northern fulmar |
| **Predicted probability change after blurring:** | Black footed albatross: | $0.221 \rightarrow 0.013$ | $0.221 \rightarrow 0.041$ | $0.221 \rightarrow 0.835$ | $0.221 \rightarrow 0.002$ |
| | Northern fulmar: | $0.731 \rightarrow 0.953$ | $0.731 \rightarrow 0.952$ | $0.731 \rightarrow 0.002$ | $0.731 \rightarrow 0.962$ |

Figure 1: Comparison between GradCAM and weighted contrastive GradCAM (proposed framework) on an image from CUB-200 dataset. The upper row contains heatmaps for two classes: black footed albatross (bfa.) and northern fulmar (nf.) The bottom row contains the blurred images according to the heatmaps (pixels with negative explanation values are blurred). Red/blue numbers indicate the changed probability with the input blurred images. The blue/red numbers are expected to decrease/increase if the explanations capture pixels with class-discriminative power.

targeted users (Preece et al., 2018) for a specific technique. As a user-oriented aspect of AI, one of the most important duties of XAI is to serve as the pipeline between the black-box models and the end-users. Therefore, it is crucial to pay attention to what end-users really need in explanations.

Here we take the classification problem as an example. when explaining a classifier, most existing explanation methods seek answers to the question "why is the sample classified as class $t$?". However, multiple studies (Lipton, 1990; Barnes, 1994; Miller, 2019) have argued that when people seek explanations of classification, they implicitly ask "why is the sample classified as class $t$, *but not others*?" Consequently, there is a mismatch between what the end users desire and what current explanation methods can deliver. Building class-contrastive explanations (viz. finding features that distinguish one class from others) plays a fundamental role in answering the question end users truly care about. However, class-contrastive explanations are less studied in literature.

To address this challenge, in this paper we propose a novel method for class-contrastive explanations. We first introduce the criterion of notations for the sake of lucid presentation. All bold lowercase letters represent tensors. And unless especially specified, $x, y, p, \phi$ represent tensors of inputs, predicted logits, probabilities and explanations, respectively. All normal lowercase letters represent scalars. Especially, $d, c, t$ represent the input dimension, output dimension, target class, respectively. Other letters will be clarified when they are used.

In machine learning, classification tasks are generally formulated as a prediction model where the output logits $y \in \mathbb{R}^c$ are the regression predictions for each individual class. Afterwards, a softmax activation (or sigmoid for binary case) is applied to compute the probabilities $p \in [0, 1]^c$ of $c$ classes. This framework is extensively applied in all kinds of classification tasks. However, for classification models, most existing back-propagation explanation methods only focus on the output of a single logit $y_t$ given a target class $t \in [c]$. This kind of explanations does **not** contain any information about other classes $s \in [c], s \neq t$. This criterion is only adequate for: i) regression tasks where the relation between $y_t$ and $y_s$ does not matter or ii) binary classification tasks ($c = 1$) when the model is itself contrastive. Otherwise, it will raise problems as features important for one class can be as equally important, or even more important for other classes. In fig. 1, GradCAM and our proposed weighted contrastive GradCAM are compared. We observe that if we focus on the top two classes bfa. and nf. separately, the original GradCAM indeed correctly highlights the object of the image. Here since $y_{\text{nf.}}$ and $y_{\text{bfa.}}$ are both relying on the object, the information provided by the explanations is limited. When features with negative explanation values are blurred according to the each class, it is expected that the probability of corresponding class should increase – or at least it shouldn't decrease much. However, it can be found that since most of the object features are important for both classes, but more important for nf., the probability of bfa. even drops when the negative features are blurred (second column). As a comparison, for the weighted contrastive GradCAM, when features with

negative explanations are blurred, the probability increases significantly. Also, since these two classes are dominant, increasing one of them automatically lead to the decay of the other one. More general experiments are in section 5. Here we demonstrate via GradCAM as it is one of the most popular explanation methods, while the above issue also exists in other back-propagation methods. On the other hand, in most perturbation methods and Integrated Gradient (Sundararajan et al., 2017), the explanations are towards the probability instead of the logit, creating explanations in a far different way from other back-propagation methods. But these methods did not explicitly distinguish the difference between explanations towards probability and logit.

Different as they are, explanations of the logit and the probability are usually obfuscated. Researchers seem to haven't noticed the difference caused by this choice. Also, since `PyTorch` classification models are usually built without the softmax activation, some methods (e.g. Integrated Gradient), even though they are proposed to explain the probability, are still widely deployed to explain the logit. Such problems can be found in popular tools such as `captum` and `torchray`. This inconsistency can be fatal when evaluating and applying explanations to open the black boxes.

Also, it should be noticed that the object area is not necessarily the "correct" area since the models may make the prediction based on unexpected features (such as the person in the top right corner). It is discovered that humans tend to evaluate probabilities only by representativeness (Kahneman et al., 1982), which means we may subjectively treat the consistency between the heatmap and the image as better explanation. Such bias is also referred to as plausibility (Jacovi and Goldberg, 2020), which is a vicious pitfall for evaluating explanations. We summarize our main contributions as follows:

- We propose the weighted contrastive scheme, a framework for existing back-propagation methods that utilizes the information of all classes to generate class-contrastive explanations.
- We validate that the proposed weighted contrastive scheme corresponds to back-propagation directly from the softmax probability.
- We demonstrate the significant difference between explanations towards logits and probabilities, and point out the fatal obfuscation in existing XAI studies.

## 2   Related Work

According to the ways explanations are produced, general post-hoc explanation methods can be roughly split into perturbations, back-propagations, and approximations. Perturbation methods, such as RISE (Petsiuk et al., 2018), extremal perturbations (Fong et al., 2019), SHAP (Lundberg and Lee, 2017) etc., try to generate explanations by purposely perturbing the input images. However, such methods usually require calling the black-box models multiple times. Generating explanation for one single input image may take up to 20 seconds, which is very inefficient in applications. Approximation methods use an external agent as the explainer for the black-box models. Such as LIME (Ribeiro et al., 2016), FLINT (Parekh et al., 2021), etc. Such explainers, as independent from the black-box models, may have trouble capturing the inner mechanism of the black boxes. Back-propagation methods make use of the back-propagation scheme to generate gradient or gradient-related explanations. Simonyan et al. (2013) propose to use input gradient with respect to some output value as the feature explanations, which are essentially the local sensitivity of the input sample. Input × Gradient (Shrikumar et al., 2016) use the Hadamard product of the input and the gradient as explanations, such that more information of the input samples is included. Integrated Gradient (Sundararajan et al., 2017) applies the gradient theorem to assign attributions to input features. GradCAM (Selvaraju et al., 2017) use the summation of CNN activations across channels weighted by the means of corresponding gradients. There are also rule-based back-propagation methods that use different ways to propagate the scores, such as LRP (Bach et al., 2015), DeepLift (Shrikumar et al., 2017), etc. Apart from general explanations, there are contrastive/counterfactual explanations that especially focus on contrastive samples, which are similar to the original sample but is predicted differently. Most contrastive explanation methods achieve this by generating contrastive samples (Dhurandhar et al., 2018; Liu et al., 2019; Agarwal and Nguyen, 2020; Jung et al., 2020; Zhao, 2020; Looveren and Klaise, 2021). Such methods are achieved by sophisticated generative models, which can be very complex. This is against Occam's razor. Wang and Vasconcelos (2020) argue that it is better to find existing samples as the contrastive ones instead of generating them. And Goyal et al. (2019) achieve this by a searching algorithm, which is less efficient.

Our work is most related to CLRP (Gu et al., 2018), where information of multiple classes are taken in to considerations. But our proposed weighted scheme properly assign weights to different

classes, leading to better contrastive results. Also, CLRP only focuses on modifying LRP, while our scheme is applicable to various back-propagation methods. SGLRP (Iwana et al., 2019) consider the influence of the softmax operation in explanations. But they use the softmax gradient only as the initial relevance, which is insufficiently justified.

# 3 Motivation

## 3.1 Sanity Check for Back-Propagation Methods

Arguably acknowledged as self-interpretable, linear models' explanations are typical attribution explanations, where each input feature $x_i$ is assigned an explanation value, $w_i^t x_i$, for a given target class $t$. Here we take linear models as a sanity check for existing criterion since attribution values of linear models are the special case of explanations of ReLU networks (Ancona et al., 2017). However, such attribution explanations are easy to manipulate. Given a target, we can completely change the score of any feature without modifying either the classification or the inner mechanism. For example, increasing $w_i^s$ (coefficient of the $i$-th feature ) for all classes $s \in [c]$ by the same amount $\delta > 0$ does not change the prediction result nor the prediction mechanism since $\boldsymbol{y}' = \boldsymbol{y} + \delta x_i \mathbf{1}$. WLOG, assume $x_i > 0$. However, the explanation value for feature $x_i$ changed by $\delta x_i$ for any target class. This is obviously not a proper way to explain classification results. Contrastively, it should be noticed that $(w_i^t - \frac{1}{c-1} \sum_{s \neq t} w_i^s) x_i$ stays invariant under the modification. The necessity of contrast here is because $x_i$ becomes important for all classes, but single-class explanations will not capture this.

## 3.2 Sigmoid v.s. Softmax

For binary classification, $c = 1$ would be sufficient. For linear model, this means $y = \boldsymbol{w}^T \boldsymbol{x} \in \mathbb{R}$ is an adequate prediction here, and $p = \frac{1}{1+e^{-y}}$ is the probability of the positive class. Here $w_i x_i$ sufficiently explains the contribution of feature $x_i$ in predicting the probability. Compared with the redundant case where $c = 2$ and $\boldsymbol{y} = \boldsymbol{W}^T \boldsymbol{x} \in \mathbb{R}^2$, neither $w_i^1 x_i$ nor $w_i^2 x_i$ is solely sufficient for explaining the classification result any more. They only explain $y_1$ and $y_2$ respectively. Intuitively, $(w_i^1 - w_i^2) x_i$ in softmax case ($c = 2$) plays the same role as $w_i x_i$ in sigmoid case ($c = 1$) for binary classification. In fact, we have the following proposition. The proof is available in the Appendix.

**Proposition 1.** *For binary classification, given the same data distribution $P(\boldsymbol{x}, u)$, $\boldsymbol{w}_1 - \boldsymbol{w}_2$ ($c = 2$, softmax) are equivalent to $\boldsymbol{w}$ ($c = 1$, sigmoid) under gradient descent optimization.*

As a result, the explanations with respect to $y$ under sigmoid activation is equivalent to the explanations with respect to $y_1 - y_2$ under (binary) softmax activation. That is, in the $c = 2$ case, we need the contrastive explanations with respect to $y_1 - y_2$ for sufficiently explaining the classifier. However, most explanation methods either just consider $y_t$ as the explanation target, which creates an inconsistency with the binary classification case, or do not distinguish explicitly the difference between $y$ and $p$.

## 3.3 Weighted Contrast across Classes

For the sake of consistent explanations, it is required to perform class-contrastive explanations even for multi-class classification tasks. For the sake of sanity check, we revisit the linear model $\boldsymbol{y} = \boldsymbol{W}^T \boldsymbol{x}$, and let $\phi_i^t(\boldsymbol{x})$ denote the attribution score of feature $i$ towards class $t$. Then contrastive attributions are usually implemented by mean contrast (Gu et al., 2018) and max contrast (Dhurandhar et al., 2018), which, in linear case, can be summarized as follows.

- **Mean Contrast:** $\phi_i^t(\boldsymbol{x})_{\mathrm{mean}} = (w_i^t - \frac{1}{c-1} \sum_{s \neq t} w_i^s) x_i$
- **Max Contrast:** $\phi_i^t(\boldsymbol{x})_{\mathrm{max}} = (w_i^t - w_i^{s^*}) x_i$ where $s^* = \arg \max_{s \neq t} y_s$

These two schemes degenerates to $y_1 - y_2$ when $c = 2$. But both of them have information loss. Mean contrast fails when different classes have different attribution scores for the same feature $x_i$, while max contrast fail to capture the feature importance for all other $c - 2$ classes. Hence, the judicious way is to combine the two contrast strategies above by summing over all classes with a series of weights $\{\alpha_s\}_{s \neq t}$. Therefore, we propose weighted contrast as follows:

- **Weighted Contrast:** $\phi_i^t(\boldsymbol{x})_{\mathrm{weighted}} = (w_i^t - \sum_{s \neq t} \alpha_s w_i^s) x_i$, where the weights $\alpha_s = e^{y_s} / (\sum_{k \neq t} e^{y_k})$ are the softmax activation of $\boldsymbol{y}_{\backslash t} \in \mathbb{R}^{c-1}$ (the predicted logit vector without the target class $t$), and $\sum_{s \neq t} \alpha_s = 1$.

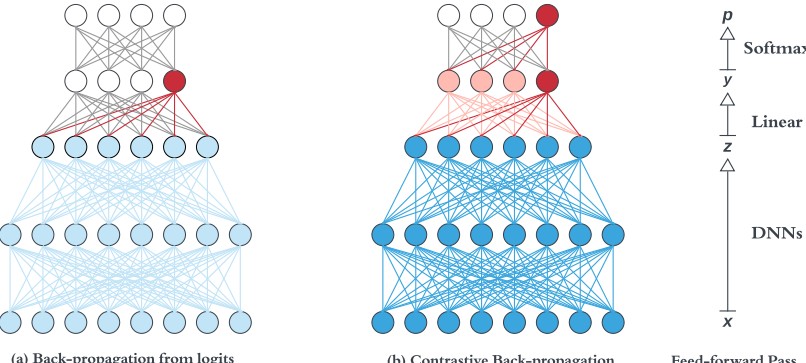

(a) Back-propagation from logits      (b) Contrastive Back-propagation      Feed-forward Pass

Figure 2: Illustration of the difference between (a) back-propagation from logits and (b) contrastive back-propagation. In the linear classification layer $z \mapsto y$, most existing methods only take one class into consideration, while contrastive methods cover all of them. A proper combination of such information is equivalent to the back-propagation from $p$.

This weighted contrast assigns class-wise importance based on the predicted logits $y_s$ of contrastive classes. In this manner, $\forall i \in [d]$, information of all $c$ classes is included (for linear model, it is the row vector $\boldsymbol{w}_i \in \mathbb{R}^c$ of the coefficient matrix $\boldsymbol{W} \in \mathbb{R}^{d \times c}$) in the attribution score of feature $x_i$. Extending the weighted contrast to a general classifier beyond linear model, we have:

$$\phi_i^t(\boldsymbol{x})_{\text{weighted}} = \phi_i^t(\boldsymbol{x}) - \sum_{s \neq t} \alpha_s \phi_i^s(\boldsymbol{x}). \tag{1}$$

**Proposition 2.** *Weighted contrastive explanations (in (1)) towards predicted logits $\boldsymbol{y}$ are equivalent to explanations towards $\boldsymbol{p} = \texttt{softmax}(\boldsymbol{y}) \in [0, 1]^c$ in gradient methods.*

In section 4, we extend proposition 2 and show the equivalence under a wide range of other back-propagation explanation methods.

### 3.4 Weighted Contrast in Back-Propagation Methods

Although linear models are constantly argued to be too simple for theoretical analysis in deep learning research, it is of more significance in back-propagations. This is because as an inductive process, back-propagation is equivalent at each step. The back-propagation steps start from the top layer to the bottom. In fact, a classification DNN $f : \boldsymbol{x} \mapsto \boldsymbol{y}$ can be decomposed as $f = f_2 \circ f_1$ where $f_1 : \boldsymbol{x} \mapsto \boldsymbol{z}$ is a DNN and $f_2 : \boldsymbol{z} \mapsto \boldsymbol{y}$ is the linear classifier. As shown in fig. 2, in $f_1$ features and learnable parameters are shared across all $c$ classes, which means that the crucial classification process is still achieved by a single linear model $f_2$. Truncated from $\boldsymbol{z}$, DNN becomes a linear model where the goal is to linearly classify $\boldsymbol{z}$ that lies in the learnt manifolds. However, when methods only back-propagate from a single $y_t$, they only utilize $1/c$ information of the linear model and the influence of losing of $(c-1)/c$ information becomes more significant as the number of classes $c$ increases. And such significant loss will all be back-propagated equally to any lower layers. We prove the following proposition in the appendix.

**Proposition 3.** *For linear back-propagation rules, the weighted contrast of a back-propagated explanation from the logit is equivalent to back-propagating the weighted contrast of original explanations of any step.*

## 4 Converting Back-Propagation Explanations to Weighted Contrast

Based on the Jacobian $J_{\boldsymbol{y}}\boldsymbol{p}$, weighted contrast scheme is applicable to various existing back-propagation-based post-hoc explanation methods. And in practice, it does not require implementing one explanation method $c$ times. We show that for most methods, it suffices to apply the original explanation schemes towards $p_t$ instead of $y_t$. Therefore, it can be performed using any existing implementations. We split back-propagation methods into two categories based on the endpoint of the back-propagation. The first class includes methods that back-propagate until the input space $\mathbb{R}^d$, such

methods include Gradient, Input × Gradient, Integrated Gradient, Layerwise Relevance Propagation, etc. The second class includes methods that back-propagate only to activations of some CNN layers, such as GradCAM, Linear Approximation (LA).

**Gradient** As the most commonly used attribution explanation method, the gradient $\phi^t(\boldsymbol{x})_y = \nabla_{\boldsymbol{x}} y_t$ captures local sensitivity of $y_t = f_t(\boldsymbol{x})$ with respect to the change of $\boldsymbol{x}$. However, even $t$ being the target class, the change of $y_t$ is not decisive to the classification result. It is the change of the relative relations between $y_t$ and $\forall s \in [c], s \neq t, y_s$, that decide the result. Note that $\phi^t(\boldsymbol{x})_{\text{weighted}}$ is equivalent to $\phi^t(\boldsymbol{x})_p$, and hence is the sensitivity of the predicted probability with respect to the input, which truly illustrates the contrastive features.

**Input × Gradient (IxG)** Since IxG is the direct Hadamard product between the input and the gradient, that is, $\phi^t(\boldsymbol{x})_y = \boldsymbol{x} \odot \nabla_{\boldsymbol{x}} y_t$. The weighted contrastive IxG is therefore defined as $\phi^t(\boldsymbol{x})_{\text{weighted}} = \phi^t(\boldsymbol{x})_y - \sum_{s \neq t} \alpha_s \phi^s(\boldsymbol{x})_y = \boldsymbol{x} \odot (\nabla_{\boldsymbol{x}} y_t - \sum_{s \neq t} \alpha_s \nabla_{\boldsymbol{x}} y_s)$, which is equivalent to IxG explanation towards $\boldsymbol{p}$.

**Integrated Gradient (IG)** As an application of the gradient theorem, IG requires computing the integral $\int_0^1 \frac{\partial f}{\partial x_i}(\tau \boldsymbol{x}) \mathrm{d}\tau$, whose closed form is infeasible. (Hesse et al., 2021) show that for ReLU DNNs without bias terms, the model $f$ degenerates to a homogeneous function, and thereby $\int_0^1 \frac{\partial f}{\partial x_i}(\tau \boldsymbol{x}) \mathrm{d}\tau = \frac{\partial f(\boldsymbol{x})}{\partial x_i}$. Therefore, IG with zeros baselines are equivalent to the IxG method.

**Rule-Based Methods** Different from back-propagation methods that rely on natural gradients, there are rule-based methods that slightly modify the back-propagation process. Such as Layerwise Relevance Propagation (LRP), DeepLift (DL), etc. Due to the special structure of DNNs, such methods actually correspond to natural-gradient methods. LRP is equivalent to IxG in ReLU networks, and LRP, DL, IxG, IG are all equivalent under homogeneous networks (e.g. ReLU/LeakyReLU networks with no bias terms). Besides, they have implementation drawbacks as constraints on modules. Moreover, such methods require manually implementing the back-propagation instead of obtaining from the built-in gradient module. As a result, their implementation time is longer than general back-propagation method.

**GradCAM** Suppose $a_{ij}^k$ to be the output of the CNN layer, where $0 < i, j < d'$ are the spatial coordinates, and $k$ is the channel. Then for GradCAM the explanation of class $t$ is

$$\phi^t(\boldsymbol{x})_y = \sum_k \left( \frac{1}{d'^2} \sum_{i,j} \frac{\partial y_t}{\partial a_{ij}^k} \right) \boldsymbol{a}^k \tag{2}$$

where we remove the last ReLU operation because the negative areas are treated as contrastive features (Selvaraju et al., 2017). And the GradCAM explanation to $p_t$ is

$$\phi^t(\boldsymbol{x})_p = \sum_k \left( \frac{1}{d'^2} \sum_{i,j} \sum_{s \in [c]} \frac{\partial p_t}{\partial y_s} \frac{\partial y_s}{\partial a_{ij}^k} \right) \boldsymbol{a}^k = \sum_{s \in [c]} \frac{\partial p_t}{\partial y_s} \phi^s(\boldsymbol{x})_y \propto \phi^t(\boldsymbol{x})_{\text{weighted}} \tag{3}$$

**Linear Approximation (LA)** Different from GradCAM, LA does not use the mean value of global average pooling of the activation gradients as the weight for each channel. Instead, it calculates the activation × gradient directly. Note that this is also equivalent to LRP with respect to the activation $\boldsymbol{a}$ instead of the input $\boldsymbol{x}$. Following the same notations, $\phi^t(\boldsymbol{x})_y = \sum_k \boldsymbol{a}^k \odot \nabla_{\boldsymbol{a}^k} y_t$. Then the weighted contrastive is equivalent to back-propagation from $p_t$ since

$$\phi^t(\boldsymbol{x})_p = \sum_k \boldsymbol{a}^k \odot \left( \sum_{s \in [c]} \frac{\partial p_t}{\partial y_s} \nabla_{\boldsymbol{a}^k} y_s \right) = \sum_{s \in [c]} \frac{\partial p_t}{\partial y_s} \sum_k \boldsymbol{a}^k \odot \nabla_{\boldsymbol{a}^k} y_s \propto \phi^t(\boldsymbol{x})_{\text{weighted}} \tag{4}$$

## 5 Experiments

In this section, we perform various experiments to demonstrate the applications of weighted contrastive explanations. Based on the two genres of back-propagation methods that we discussed in section 4, we devise different experiments to validate the advantages of using weighted contrastive back-propagation explanations. All experiments are implemented through `Pytorch` on Intel Core i9-9960X CPU @ 3.10GHz with Quadro RTX 6000 GPUs.

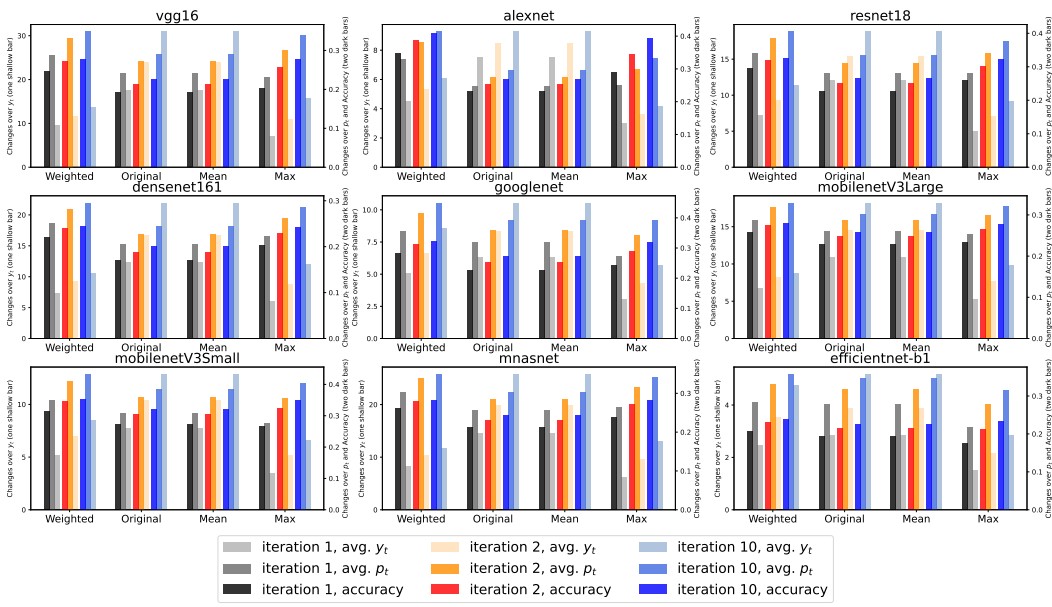

Figure 3: Changes in accuracy, $y_t$ and $p_t$ ($t$ is the target classification class) when certain input features are perturbed. The perturbed features are selected based on four gradient explanations (original, mean, max, weighted contrast), where "original" means explanations w.r.t. the logit directly. We show the results of 9 commonly used CNNs with varying iteration numbers (iteration = 1,2,10) on the ILSVRC2012 dataset. The lighter-colored bars represent the average changes over $y_t$ (right $y$-axis), and the two darker-colored bars represent the changes over $p_t$ and accuracy (left $y$-axis). The results show: 1) the changes of $p_t$ and accuracy match each other, but changes in $y_t$ do not. This is a support for using contrastive explanations; 2) our method shows higher $p_t$ and accuracy in all settings, indicating ours captures important features leading to the largest change in performance.

## 5.1 Back-Propagation till the Input Space

It is recently discovered that explanation saliency maps that are over the original space incline to highlight the textures/silhouette of the input images (Adebayo et al., 2018). As a result, visual evaluations of such saliency maps are easy to be biased by the representativeness – the more accurate the saliency map shows the original object, the better we humans might think it be. But in this case a heuristic silhouette detector will ace this test. Besides, since the information of the original input image is dominant in the saliency map, changes in the qualitative results with visualizations are almost invisible because of the scale differences. Therefore, we focus on quantitative results for such methods in the main paper. We leave the visualizations results and detailed descriptions in Appendix.

As all such methods are based or partially based on the input gradient, we propose to compare the recourse application of the weighted contrastive gradient (Weighted), original gradient (Original), mean contrastive gradient (Mean), and max contrastive gradient (Max) methods by performing the input modification based on the gradient sign perturbation over ILSVRC2012 (Deng et al., 2009) validation set, where 50000 images from 1000 classes are included. The gradient sign perturbations are implemented following the projected gradient descent (Madry et al., 2017)

$$\boldsymbol{x}^{n+1} \leftarrow \boldsymbol{x}^n + \alpha \, \mathtt{sign}(\boldsymbol{\phi}^t(\boldsymbol{x}))$$
$$\boldsymbol{x}^{n+1} \leftarrow \mathtt{clamp}(\boldsymbol{x}^{n+1}, \min(\boldsymbol{x} - \epsilon, 0), \max(\boldsymbol{x} + \epsilon, 1))$$

where we set $n \in [1, 2, 10]$ to be the number of iterations, $\epsilon = 10^{-3}$ is the perturbation limit, and $\alpha = \epsilon/n$ is the step size. 9 commonly used CNNs are tested. The results are shown in fig. 3. For each model, gray-ish, red-ish and blue-ish bars represent results of 1, 2, 10 iterations respectively. For a given step, the wide shallow bar represents the change in $y_t$, the thin median bar represents the change in $p_t$, and the thin deep bar represents the change in accuracy. It can be found that weighted contrast outperforms all other methods in improving the probability of the target class and the accuracy. And

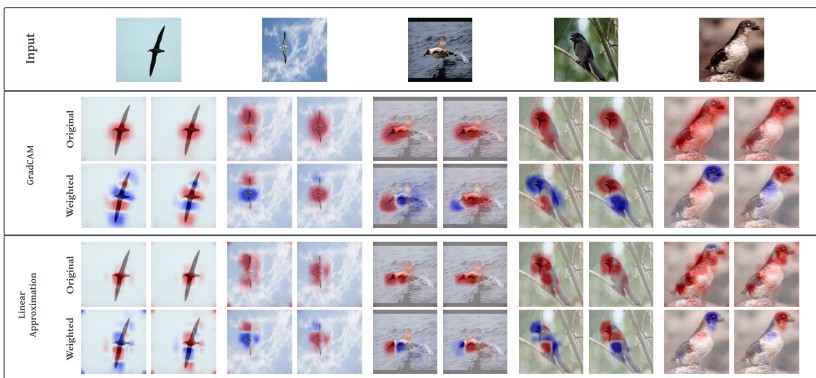

Figure 4: Comparison between the back-propagation from logits (Original) and weighted contrastive back-propagation (Weighted) over GradCAM and LA on images from the CUB-200 dataset. For each image, the top 4 images are from GradCAM and the bottom 4 images are from LA. The left/right column are explanations to the most/second possible class, respectively.

these two measures actually correspond with each other. On the contrary, the change in $y_t$ does not match the change in $p_t$ or accuracy, since higher $y_t$ does not guarantee higher accuracy or $p_t$.

## 5.2 Back-Propagation till the Activation Space

For the methods that back-propagate to activation space, the activation is set up as the last CNN layer. Due to the spatial invariance property of CNNs, the areas of activations are directly (but in an unknown manner) related to the corresponding areas of the original input. But without being biased by input silhouettes and textures, such methods' heatmaps usually demonstrate better localization power. Since one of the main goals of contrastive explanations is seeking fine-grained distinctions between features (Wang and Vasconcelos, 2020), here we carry out experiments over 5 fine-grained datasets: CUB-200 (Wah et al., 2011) containing 200 classes of birds, Fine-Grained Visual Classification of Aircraft (FGVC) (Blaschko et al., 2012) containing 100 classes of aircraft, Food-101 (Bossard et al., 2014) containing 101 classes of food, Flower-102 (Nilsback and Zisserman, 2008) containing 102 classes of flowers, and Stanford Cars (Krause et al., 2013) containing 196 classes of cars. We test the results on VGG-16 (Simonyan and Zisserman, 2014) and AlexNet (Krizhevsky, 2014) that are pretrained by `PyTorch` on ImageNet (Deng et al., 2009) and fine-tuned correspondingly.

Here we mainly focus on the contrastive explanations of the top 2 classes with the highest predicted probabilities, which conforms to what humans usually pay attention to in classification tasks. For example, people are usually interested in understanding why the model does not classify `apple_pie` as `bread_pudding`, rather than why does the model not classifying `apple_pie` as `greek_salad`. We clarify that our method is not limited to explaining 2 classes. In fact, our method can provide contrastive explanations with respect to any subset of classes of interest.

### 5.2.1 Visualizations

GradCAM and Linear Approximation results under the original form and the proposed scheme are shown in fig. 4. Images are selected so that the second possible class has probability higher than 0.1. Comparing the two columns of each input image, we can find that for the two most possible classes, the original methods back-propagates from the logits, which will always give very similar explanations, because two explanations are independent. Such explanation pairs convey limited information about the classification to the audience. On the contrary, weighted methods have focus on specific parts in explanations of different classes. Also, although the weighted explanations of the top 2 classes seem to be complementary, they do not necessarily need to be. This phenomenon is because these two classes are dominant over all $c$ classes. We also present very detailed analysis of the five bird images in appendix D Visualizations of other datasets can be found in appendix E.

### 5.2.2 Blurring/Masking

We perform blurring experiments, where we blur positive and negative features decided by original and weighted GradCAM/LA, respectively. Since CNNs take input of a fixed size, missingness is

Table 1: Comparisons between weighted contrastive method (wtd.) and original method (ori.) in blurring/masking experiments on 5 datasets. 3 baselines (Gaussian Blur, Zeros, Channel-wise Mean), and 2 methods (GradCAM, LA) are included. $t_1, t_2$ represent the classes with the highest and the second highest probability respectively. In each line, we show the changes of expected relative probability. Pos./Neg. Features mean that only positive/negative features are kept with respect to the corresponding $t_i$ class. It is expected that when the positive/negative features corresponding to $t_i$ are kept, the expected relative probability $\bar{p}_i$ is expected to increase/decrease.

| | | | | Gaussian Blur | | | | Zeros | | | | Channel-wise Mean | | | |
| | | | | Pos. Features | | Neg. Features | | Pos. Features | | Neg. Features | | Pos. Features | | Neg. Features | |
| | | | $p_t$ | ori. | wtd. | ori. | wtd. | ori. | wtd. | ori. | wtd. | ori. | wtd. | ori. | wtd. |
|---|---|---|---|---|---|---|---|---|---|---|---|---|---|---|---|
| CUB-200 | LA | $t_1$ | 0.710 | 0.753 | **0.820** | 0.447 | **0.280** | 0.730 | **0.795** | 0.464 | **0.301** | 0.741 | **0.815** | 0.457 | **0.294** |
| | | $t_2$ | 0.290 | 0.450 | **0.749** | 0.417 | **0.178** | 0.447 | **0.723** | 0.424 | **0.205** | 0.438 | **0.741** | 0.410 | **0.187** |
| | GC | $t_1$ | 0.710 | 0.752 | **0.845** | 0.440 | **0.248** | 0.740 | **0.830** | 0.444 | **0.259** | 0.747 | **0.844** | 0.446 | **0.250** |
| | | $t_2$ | 0.290 | 0.406 | **0.776** | 0.418 | **0.137** | 0.410 | **0.762** | 0.416 | **0.152** | 0.403 | **0.773** | 0.409 | **0.141** |
| FGVC | LA | $t_1$ | 0.693 | 0.757 | **0.808** | 0.397 | **0.248** | 0.718 | **0.742** | 0.435 | **0.312** | 0.736 | **0.772** | 0.415 | **0.287** |
| | | $t_2$ | 0.307 | 0.627 | **0.775** | 0.377 | **0.193** | 0.606 | **0.711** | 0.417 | **0.261** | 0.614 | **0.742** | 0.404 | **0.230** |
| | GC | $t_1$ | 0.693 | 0.792 | **0.862** | 0.388 | **0.192** | 0.757 | **0.813** | 0.421 | **0.256** | 0.769 | **0.835** | 0.405 | **0.231** |
| | | $t_2$ | 0.307 | 0.617 | **0.826** | 0.352 | **0.145** | 0.602 | **0.763** | 0.392 | **0.203** | 0.606 | **0.787** | 0.379 | **0.179** |
| Food-101 | LA | $t_1$ | 0.714 | 0.805 | **0.872** | 0.344 | **0.176** | 0.793 | **0.836** | 0.375 | **0.215** | 0.797 | **0.848** | 0.361 | **0.200** |
| | | $t_2$ | 0.286 | 0.696 | **0.838** | 0.322 | **0.126** | 0.684 | **0.803** | 0.352 | **0.164** | 0.689 | **0.817** | 0.348 | **0.150** |
| | GC | $t_1$ | 0.714 | 0.828 | **0.907** | 0.339 | **0.128** | 0.821 | **0.891** | 0.353 | **0.151** | 0.824 | **0.896** | 0.342 | **0.139** |
| | | $t_2$ | 0.286 | 0.669 | **0.888** | 0.317 | **0.085** | 0.668 | **0.865** | 0.334 | **0.106** | 0.669 | **0.874** | 0.328 | **0.098** |
| Flower-102 | LA | $t_1$ | 0.715 | 0.778 | **0.869** | 0.423 | **0.290** | 0.700 | **0.776** | 0.350 | **0.268** | 0.719 | **0.785** | 0.390 | **0.274** |
| | | $t_2$ | 0.285 | 0.598 | **0.746** | 0.386 | **0.133** | 0.632 | **0.744** | 0.450 | **0.234** | 0.634 | **0.736** | 0.448 | **0.223** |
| | GC | $t_1$ | 0.715 | 0.796 | **0.870** | 0.378 | **0.217** | 0.774 | **0.858** | 0.345 | **0.177** | 0.777 | **0.868** | 0.370 | **0.194** |
| | | $t_2$ | 0.285 | 0.539 | **0.805** | 0.369 | **0.110** | 0.543 | **0.835** | 0.416 | **0.170** | 0.544 | **0.827** | 0.411 | **0.164** |
| Stanford Cars | LA | $t_1$ | 0.721 | 0.777 | **0.835** | 0.426 | **0.259** | 0.774 | **0.830** | 0.437 | **0.263** | 0.772 | **0.836** | 0.438 | **0.260** |
| | | $t_2$ | 0.279 | 0.574 | **0.765** | 0.412 | **0.156** | 0.589 | **0.762** | 0.422 | **0.162** | 0.592 | **0.769** | 0.412 | **0.159** |
| | GC | $t_1$ | 0.721 | 0.810 | **0.899** | 0.416 | **0.199** | 0.803 | **0.894** | 0.421 | **0.206** | 0.804 | **0.898** | 0.419 | **0.205** |
| | | $t_2$ | 0.279 | 0.552 | **0.830** | 0.405 | **0.102** | 0.565 | **0.829** | 0.407 | **0.108** | 0.564 | **0.829** | 0.402 | **0.108** |

not a well-defined concept and is achieved by removing original pixel values and imputing with heuristic baseline values. Sturmfels et al. (2020) demonstrate that the baseline values have non-negligible impact on the results. Hence we include three different baselines respectively: 1) missing pixels are blurred; 2) missing pixels are imputed with zeros; 3) missing pixels are imputed with the (RGB) channel-wise mean values. Here VGG-16 is applied as the explained classifier, and heatmaps $14 \times 14$ are generated at the output of the CNN layers but before the last MaxPooling layer for higher resolutions. Then they are upsampled to the input space $224 \times 224$ by bilinear upsampling. The results are shown in table 1, we present the relative probability $\bar{p}_{t_i} = \mathbb{E}[e^{y_{t_i}}/(e^{y_{t_1}} + e^{y_{t_2}})], i = 1, 2$, where $t_i \in [c]$ represent the $i$-th possible class. Expectations are calculated over samples where $p_{t_2} > \hat{p}$. $\hat{p}$ is a threshold we fixed to be 0.1 here to ensure that the $t_1$ class not to be too dominant. It can be found clearly that weighted methods outperform original methods in altering the probability. This result has verified the fact that back-propagation explanations from the logits fail to capture the difference between exclusive features and features shared among classes. In contrast, our weighted contrast method effectively captures positive and negative features for both $t_1$ and $t_2$ classes with discriminative power. It should be noticed that here all samples are covered in the experiment thus the results shown in table 1 are deterministic and there is no uncertainty present table 1.

## 5.3 Comparison with Mean/Max Contrast

Recall that mean contrast and max contrast have been applied to generate contrastive explanations. We compare the explanation results between the proposed weighted contrast and these two schemes. As discussed in section 3.3, both mean and max contrast schemes have inevitable information loss in explaining predictions of different classes. As shown in fig. 5, for the mean contrastive, since the explanation is obtained by $\phi_{\text{mean}}^t = \phi^t - \frac{1}{c-1} \sum_{s \neq t} \phi^s$, where the $c-1$ non-target classes are treated equally. The classes with positive explanations and classes negative explanations are neutralized with each other by the simple summation. The denominator $c - 1 = 199$ also decreases the contrast term a lot. As a result, we can find that the mean contrast is almost the same as the original GradCAM. As for the max contrastive scheme, it **only** cares for the max non-target class, which is insufficient. As shown in the red boxes, we can find that the max contrastive explanations for these two classes are simply the additive inverse of each other. This is because when targeting one of them, the other automatically becomes "max non-target". Compared with the weighted contrastive explanations, it does not contain any information of the 3rd-possible class (third row). For instance, in the top left

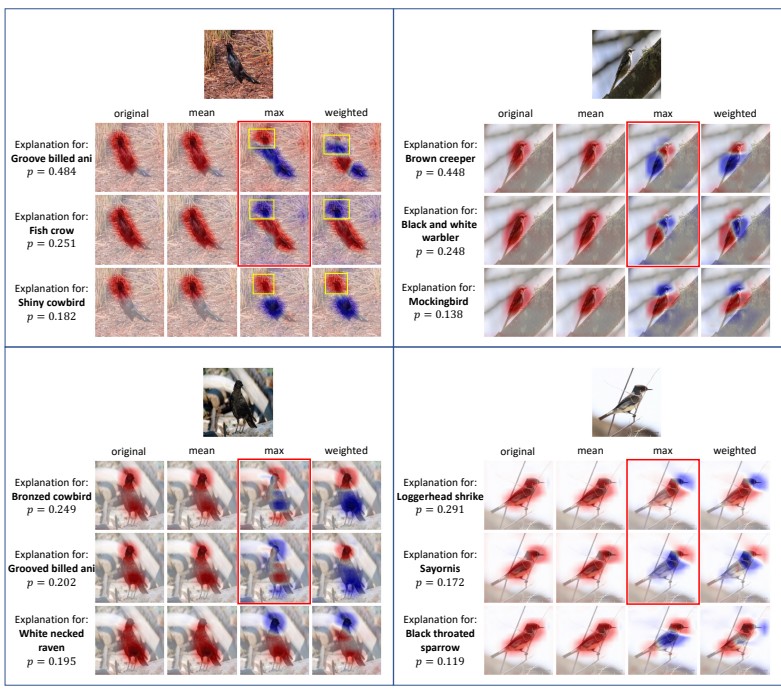

Figure 5: Comparison with mean/max contrast. 4 images from CUB-200 are presented. For each image, we present the explanations for the top 3 classes predicted by the classifier in three rows in the top-down order. And in each column, explanations from the origin, mean contrastive, max contrastive and weighted contrastive GradCAM are presented.

block, when the head of the bird contributes both to the 1st-possible class "groove billed ani" and the 3rd-possible class "shiny cowbird" , the weighted contrastive explanation manages to find out that it contributes more to "shiny cowbird". However, due to the absence of information from the 3rd-possible class, the max contrastive explanation fails to find this difference. Such explanations are insufficient to help understanding the behaviors of the black-box classifier.

# 6 Conclusions & Future Work

In this work, we propose a weighted contrastive scheme for back-propagation explanations that make proper use of information from all classes. The proposed scheme is applicable to a wide range of existing back-propagation methods and does not require additional explainers. We prove that the proposed weighted contrast, though include all $c$ classes, is actually equivalent to directly explaining the probability (softmax activation), which only requires one back-propagation for one sample. This makes it easy to implement compared with existing contrastive explanation methods. We also demonstrate the significant difference between explanations to the logit $y$ and to the probability $p$ and point out that it is important to distinguish them. Our experiments show that $y$-based explanations have better localization capability, while $p$-based explanations focus more on the contrast among classes. This should be decided based on the desiderata of explanations case by case.

Albeit the desired results of the weighted contrast, we admit that, due to the simplicity of our model, the form of explanations are still limited to the heatmaps of the input sample. But it is worth noticing the here the heatmaps can provide guidelines for generating contrastive explanations in complex models, since the highlighted areas are verified to be directly related to the classification results. We will explore the role of our weighted contrastive explanations in generating/searching for contrastive samples in future work.

## Acknowledgements

This work was partially supported by NSF IIS #1955890, IIS #2146091, Purdue's Elmore ECE Emerging Frontiers Center.

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
