# OpenReview forum: "“Why Not Other Classes?”: Towards Class-Contrastive Back-Propagation Explanations"
_NeurIPS.cc/2022/Conference — NeurIPS 2022 Accept_

### Official Review · Reviewer_PhZB · 2022-07-08

**Rating:** 4
**Confidence:** 4
**Soundness:** 1 poor
**Presentation:** 1 poor
**Contribution:** 2 fair

**Summary:**

The authors propose a method (Weighted Contrast) for explaining DNN classifier predictions. Rather than focusing on features that change the predicted probability of a given target class, the authors’ method focuses on features that are important for one class and _not_ others.

**Questions:**

See previous "Weaknesses" section.

**Limitations:**

The authors briefly discussed the limitations of their work in the conclusion section. I would strongly recommend the authors expand the limitations section and specifically discuss how backpropagating with respect to softmax/sigmoid outputs may lead to issues with some previously proposed methods (see "Weaknesses" point 3).

The authors do not discuss potential negative societal impact for this work, which I think is fine (I don't see any obvious potential for negative impacts).

**Strengths And Weaknesses:**

Strengths:

1. As shown in previous work and by the authors, so-called contrastive explanations can better capture features that discriminate between classes for a given model as compared to non-constrastive explanations.
2. The method proposed by the authors is an extension of gradient-based explanation methods, making it faster than some previously proposed contrastive explanation methods.

Weaknesses (listed in order of importance to my score):

1. Lack of baselines: Many recent works have focused on developing contrastive explanation methods. Although the authors do discuss some of these in their literature review, they do not compare their proposed method against any of these previous works in their experiments; instead, the previously proposed methods are dismissed as being "against Occam's razor". While I acknowledge the authors' point that their proposed method may be computationally cheaper than previously proposed methods, it's necessary to understand whether this speed comes at the expense of explanation quality. For example, with non-contrastive explanations, simpler gradient-based approaches have been shown to produce worse quality explanations than the axiomatically justified (but computationally more intensive) integrated gradients or SHAP methods. Without any comparisons against previously proposed methods, it is difficult to assess the significance of the authors' results.
2. Novelty/additional discussion on the choice of backpropagating with respect to $p$ or $y$: The authors' show that a "weighted contrastive explanation" of logits $y$ is equivalent to a standard explanation of softmax/sigmoid outputs $p$. However, to my knowledge backpropagating with respect to $p$ is already standard practice for some previously proposed methods (see e.g. the official Tensorflow integrated gradients tutorial at https://www.tensorflow.org/tutorials/interpretability/integrated_gradients). Given this, could the authors clarify what exactly the contribution of this work is? Is it just the theoretical perspective suggesting that one should _always_ backpropagate with respect to softmax/sigmoid outputs as opposed to logits or something more? Moreover, for other methods (e.g. input x gradients/LRP) it is specifically recommended _not_ to backpropagate with respect to softmax/sigmoid outputs (see [1] Sections 2.4/2.5). Could the authors comment on this?
3. Presentation of experimental results: In addition to point (1), I found it difficult to assess the authors' experimental results due to issues with figure design. Specifically, for the bar charts in Figure 3, the two darker-colored bars are superimposed on the lighter-colored one for each group of bars, making the chart difficult to read. Moreover, I found Figure 4 hard to evaluate without the labels of the most and second-most likely classes.
4. Writing clarity: I found it difficult to understand the authors' point in parts of the manuscript due to issues with writing clarity. Specifically, Section 3 felt quite rushed, and I had to read the section multiple times to connect the propositions with the preceding text.

[1]: "Not Just A Black Box: Learning Important Features Through Propagating Activation Differences" https://arxiv.org/abs/1605.01713

---

> ### Author Response · Authors · 2022-08-02
> **Response to Reviewer PhZB**
>
> **Comparison with baselines.**
> Thanks. We answer this from three aspects. **First**, the goals of our weighted contrastive scheme and existing contrastive/counterfactual explanation methods are different. We answer “why the instance is classified into this class, *but not others*?“. While most existing contrastive/counterfactual explanation methods answer “how to change the instance to other classes?” via generative models (Line 99-101). Due to different goals, the previous contrastive/counterfactual explanations are not proper baselines. Instead, the most related baselines are standard explanation methods (e.g., GradCAM, Linear Approximation) and we show our method improves them. **Second**, our scheme can handle more complex data (e.g., CUB-200, Food-101). But existing contrastive/counterfactual explanations are limited to preliminary data like MNIST. **Third**, per the reviewer’s suggestion, we adapt our method to compare with CEM [Dhurandhar et al., NeurIPS 2018]. CEM is the most similar to ours among existing contrastive/counterfactual explanations and one of the most cited. It’s worth noting that based on Eq. (1)(2) of [Dhurandhar et al., NeurIPS 2018], CEM is essentially a perturbation method focusing on **max contrast** (we discussed our difference from max contrast in Sec 3.3 of our paper). Due to CEM’s complexity constraint, we implement both methods on MNIST data and show results in [this figure](https://drive.google.com/file/d/1YUzaEZniS5XDtF4h4bFxVDPjkXIMt5TJ/view?usp=sharing) and the table below. The results show that applying our method improves CEM by a large margin.
>
> ||Original|Standard CEM|Weighted Contrastive
> -|-|-|-
> $p_t$|9.278e-1|3.198e-3|**1.353e-5**
>
> **Table:** Comparison of $p_t$: the average probability of predicted classes of original images vs. after perturbation by CEM/our method on the first 1,000 validation samples in MNIST. Lower values show better perturbation results.
>
> **Choices of $p$ and $y$.**
> Thanks. The discussion ([Shrikumar et al., ICML 2017] Sec 2.4/2.5) referred by the reviewer is actually a support for contrastive explanations. The reason why back-propagation (BP) through softmax is not preferred by [Shrikumar et al., ICML 2017] is the “summation to $\delta$” property (a.k.a. completeness [Sundararajan et al., ICML 2017], local accuracy [Lundberg et al., NeurIPS 2017], etc.). This property requires the summation of all feature contributions equal to (output$-$baseline). In this sense BP through softmax will cause “attenuation of contribution” because the summation of contributions is constrained to $p_t<1$. This is not an issue with the our scheme for the following reasons. (Here we omit all $\mathbf{x}$ for brevity).
> 1. Given class $t$, the we use the weighted summation of $\phi^s\_y,\forall s\in[c]$ to calculate $\phi^t\_\mathrm{weighted}$. As shown in Sec 4, this is actually proportional to $\phi^s\_p$:
> $$
> \phi^t\_{\mathrm{weighted}} = \frac{(\sum_{j=1}^ce^{y_j})^2}{e^{y_t}(\sum_{j\ne t}e^{y_j})}\cdot\phi^t\_p \simeq \phi^t\_p
> $$
> The scale factor adjusts for the attenuation caused by the summation to $\delta$ property.
> 2. The importance of contribution to different classes is also claimed in [Shrikumar et al., ICML 2017]. In Sec 2.5, Eq (25), they propose a workaround for softmax as (following the original notations)
> $$
> \bar{w}\_{xy_i} = w_{xy_i} - \frac{1}{n}\sum_{i’=1}^nw_{xy_{i’}}
> $$
> which is exactly the “Mean Contrast” that is discussed in our paper, Sec 3.3. We validate the advantages of our weighted contrast over mean contrast using both theoretical and empirical analysis.
> Besides, we clarify that introducing the BP from $p$ is not the contribution of our work. Instead, one main contribution of our work is to distinguish between $p$ and $y$ in building and evaluating explanations, which should be payed much more attention to. In fact, almost all BP methods (e.g. Gradients, Input$\times$Gradient, GradCAM, etc.) are originally proposed to BP from $y$. In popular implementations like ``captum`` and ``torchray`` they are also implemented this way. To include adequate information of all classes, we propose the weighted summation scheme, which corresponds to the BP from $p$ for some existing methods for faster calculation. As for methods like standard IG, weighted contrast also alleviates the attenuation issue mentioned above.
>
> **Presentation of results.**
> Thanks. We’ll rearrange the bar charts accordingly. We present an improved demonstration of the top left subfigure (vgg16) of Fig. 3 in our main paper [here](https://drive.google.com/file/d/1hUCTmDQ1zVBur5s2QTU6XupQsvPDHXIs/view?usp=sharing).
> For Fig. 4, thanks for the suggestion. We present a more detailed version in [this figure](https://drive.google.com/file/d/1GYcE63zUmGw8oFY1vjKHwbx3RIPguu1f/view?usp=sharing) to show weighted contrastive scheme captures contrastive features.
>
> **Writings.**
> Thanks. We’ll revise Sec 3 carefully.

---

> > ### Comment · Reviewer_PhZB · 2022-08-05
> > **Thank you for the additional work**
> >
> > Dear authors,
> >
> > Thank you for providing the additional results comparing with CEM and for clarifying the point on backpropagation through softmax <->  [Shrikumar et al., ICML 2017]. In light of your rebuttal I have increased my score.
> >
> > However, at this time I still cannot recommend acceptance, as I still have concerns over the novelty/presentation of backpropagating with respect to $p$ vs $y$. To my knowledge, your claim [1] (which was repeated in the rebuttal) on previous work proposing to specifically backpropagate with respect to $y$ is **incorrect**. For example, in the manuscript introducing integrated gradients [2], the authors in Definition 1 define the neural network that they wish to probe as a function $F \colon R^n \to [0, 1]$ (i.e., they are _not_ considering unnormalized logits). This is further reflected in the code repository [3] accompanying the paper, in which in integrated gradients is applied to softmax normalized probabilities. Similarly, in the `captum` repository [4] mentioned in the authors' rebuttal, the provided example for integrated gradients assumes that attributions are computed with respect to probabilities and not unnormalized logits.
> >
> > To be clear, I think that the authors have demonstrated that attributing with respect to softmax-normalized probabilities is superior to that of unnormalized logits, and this is a nice contribution. However, the manuscript and rebuttal are framed as if attributing with respect to unnormalized logits is standard practice when, in fact, attributing with respect to softmax-normalized probabilities is already the norm.
> >
> > [1]: Lines 54-56 of the manuscript: "However, for classification models, most existing back-propagation explanation methods only focus on the output of a single logit $y_t$ given a target class $t ∈ [c]$."
> >
> > [2]: Sundararajan et al., Axiomatic Attribution for Deep Networks
> >
> > [3]: https://github.com/ankurtaly/Integrated-Gradients. In particular, see https://github.com/ankurtaly/Integrated-Gradients/blob/master/InceptionModel/inception_utils.py lines 33-34, in which softmax outputs are extracted
> >
> > [4]: https://github.com/pytorch/captum/blob/master/captum/attr/_core/integrated_gradients.py see lines 255-261

---

> > > ### Author Response · Authors · 2022-08-06
> > > **Addressing Further Concerns**
> > >
> > > Dear Reviewer,
> > >
> > > Thank you for mentioning the concerns over backpropagating with respect to $p$ vs. $y$. We'd like to clarify that the difference between explanations with respect to $p$ or $y$ in explaining black boxes is vital, but lacks attention in the current Explainable Artificial Intelligence (XAI) study. We address the reviewer's concerns with the following details:
> > >
> > >
> > > 1. For Integrated Gradient (Sundararajan et al., ICML 2017), although it is officially introduced to focus on $p$ (Sec. 3, Paragraph 2, $F:\mathbb{R}^n\rightarrow[0,1]$ as mentioned by the reviewer), there's no justification on why it is chosen. And in the definition of completeness (Proposition 1), the function $F: \mathbb{R}^n\rightarrow\mathbb{R}$ is again the logit function instead of the probability function. (Otherwise the statement "it is possible to choose a baseline such that the prediction at the baseline is near zero" does not hold, because the softmax layer will force the predicted probability to near $1/$(\# of classes) for an almost-zero logit tensor). This is also reflected in a follow-up work of IG [(Hesse et al., in NeurIPS 2021)](https://proceedings.neurips.cc/paper/2021/file/a284df1155ec3e67286080500df36a9a-Paper.pdf), where IG is again defined **without** the softmax layer.
> > >
> > >
> > > 2. For ``captum``, it is an API designed to handle a general ``PyTorch`` model, and whether the explanation is with respect to $y$ or $p$ is decided by the implemented model output. General implementation of PyTorch models does not have the softmax layer built-in with it, as the ``nn.CrossEntropyLoss()`` takes y and label as input to calculate the loss. As a consequence, even though it was claimed in the [link](https://github.com/pytorch/captum/blob/master/captum/attr/_core/integrated_gradients.py) referred by the reviewer that the model "returns an Nx10 tensor of class probabilities", ``captum`` is **not** using this way. Please see this [``captum tutorial for ResNet``](https://captum.ai/tutorials/Resnet_TorchVision_Interpret) using **Integrated Gradient**. In this tutorial, Integrated Gradient is performed directly to the ``torchvision.models.resnet18`` model (in block (7)), which does **not** have a softmax activation ([Line 266-285](https://github.com/pytorch/vision/blob/main/torchvision/models/resnet.py)). This means that Integrated Gradient is used **without** the softmax layer, despite explicitly mentioned in the paper (Sundararajan et al., ICML 2017) to backpropagate from $p$. Another example by ``captum`` is [``this tutorial for CIFAR``](https://captum.ai/tutorials/CIFAR_TorchVision_Interpret), where the model ``net`` defined in block (4) is built **without** the softmax activation, and then the **Saliency (Gradient)** explanations are directly generated for ``net`` (in block (10)). However, it is still claimed in the source code [Lines 111-119](https://github.com/pytorch/captum/blob/master/captum/attr/_core/saliency.py) that **Saliency** should explain the model that outputs $p$.
> > >
> > > Considering these inconsistent details, it is apparent that currently focusing on $p$ or $y$ is more like a casual choice rather than a careful decision in explanation method design, implementation, or use of the explanations, which can lead to chaos in the XAI research.
> > >
> > > As we mentioned before, our contribution is not to come up with the idea of focusing on $p$, as it is simply shifting the target and already be done by some work such as Integrated Gradient. Instead, by introducing the **weighted contrast**, we point out clearly that focusing on $p$ and $y$ are two distinct procedures and will result in profoundly different explanations. Such difference, to our best knowledge, has not been studied before. Besides, it will lead to many valuable discussions, such as 1) Is it valid to evaluate explanation methods that explain different targets? 2) Now that so many explanations are focusing on $y$, is it appropriate to evaluate them using $p$ or even accuracy? 3) Among all the existing comparisons in the existing work, how exactly are these explanations implemented as the benchmark? 4) What questions do we want the explanations to answer when we propose an explanation? etc. Note that we are not claiming the overall superiority of weighted contrast or BP from $p$ over BP from $y$. They are answering different questions. As a result, explaining $y$ and $p$ should be carefully studied and distinguished, rather than in the current chaotic way.

---

### Official Review · Reviewer_aXtB · 2022-07-11

**Rating:** 6
**Confidence:** 3
**Soundness:** 3 good
**Presentation:** 3 good
**Contribution:** 3 good

**Summary:**

In this paper, the authors argued for the need of "class-contrastive" explanations, which not only explain "why an input is classified into a particular class," but also "why the input is NOT classified as others." To obtain class-contrastive explanations, the authors proposed the "weighted contrast" explanation scheme, which is similar to the previous "mean contrast" explanation scheme except for the weights assigned to the non-target classes -- namely, in "mean contrast" explanation scheme all non-target classes have the same weight, whereas in the proposed "weighted contrast" explanation scheme, the weight for a particular non-target class depends on the softmax normalized score for that non-target class against the scores for all non-target classes. The authors showed that their proposed weighted contrastive explanation scheme is equivalent to standard explanation applied to the softmax normalized output of the model, thereby simplifying the implementation of their explanation scheme. Finally, they applied their weighted contrastive method to various back-propagation explanation techniques, including gradient, input x gradient, integrated gradient, GradCAM, and linear approximation (LA), and showed quantitatively that their weighted contrastive explanation scheme can find input regions that are better associated with target-class probabilities, and also qualitatively that their weighted contrastive explanation scheme (when applied to GradCAM and LA) provides more informative explanations.

**Questions:**

See the "weaknesses" above -- I have questions regarding the validity of the last part of equations (3) and (4), as well as the validity of the claim that integrated gradient with zero baselines is equivalent to the input x gradient method. Please provide mathematical proofs of both statements.

**Limitations:**

The authors have adequately addressed the limitation that their contrastive explanation method is only applicable to techniques where attributions ("heatmaps") are involved.

**Strengths And Weaknesses:**

Strengths:

+ The authors have provided very solid theoretical reasoning for where the current explanation techniques fall short and why we need "class-contrastive" explanations.

+ The authors have provided an easy way to obtain "class-contrastive" explanations -- namely, applying standard techniques not to the logits (unnormalized class scores) but to the predicted probabilities (after softmax normalization).

+ The quantitative measurement of changes in predicted probabilities by perturbing/blurring/masking pixels or input regions identified as important does show that the proposed "class-contrastive" explanation scheme is superior in finding input regions that are relevant for a target class.

Weaknesses:

- I find it difficult to understand equations (3) and (4) -- in particular, why is the weighted sum of standard GradCAM/LA explanations for all classes (where the weight for a class is the derivative of the target class probability with respect to the unnormalized score of that class) "approximately equal" to the weighted contrastive explanation? Please show your proof.

- I also find it difficult to see how integrated gradient with zero baselines is equivalent to the input x gradient method. Please show your proof.

A minor issue regarding related work:

- ProtoPNet (Chen et al., 2019) only provides similar examples but not contrastive examples. In addition, ProtoPNet does not require additional annotations. In fact, ProtoPNet (and a number of other works) does not belong to the "posthoc" explanation family.

---

> ### Author Response · Authors · 2022-08-02
> **Response to Reviewer aXtB**
>
> We appreciate the acknowledgment of our work. All the questions and concerns are answered below.
>
> **Elaboration on Eq. (3)(4).**
> As proposed in Eq. (1), the weighted contrastive explanations for the $i$-th input feature to class $t$ have the form $\phi_i^t(\mathbf{x})\_\mathrm{weighted} = \phi_i^t(\mathbf{x}) - \sum_{s\ne t}\alpha_s\phi_i^s(\mathbf{x})$, or in the compact form
> $$
>     \phi^t(\mathbf{x})\_\mathrm{weighted} = \phi^t(\mathbf{x}) - \sum_{s\ne t}\alpha_s\phi^s(\mathbf{x})
> $$
> Here $\alpha_s = e^{y_s}/(\sum_{j\ne t}e^{y_j})$ are the weights.
>
>
> On the other hand. In Eq. (3), the GradCAM explanation to $p_t$ is
> $$
>     \phi^t(\mathbf{x})\_p = \sum_{s=1}^c\frac{\partial p_t}{\partial y_s}\phi^s(\mathbf{x})
> $$
> where $c$ is the number of classes, and $\phi^s(\mathbf{x})$ is the GradCAM explanation to $y_s$.
> Notice that $\mathbf{p}$ is the softmax activation of $\mathbf{y}$, we have
> $$
>     \frac{\partial p_t}{\partial y_t} = \frac{e^{y_t}(\sum_{j\ne t}e^{y_j})}{(\sum_{j=1}^ce^{y_j})^2} = \biggr[\frac{e^{y_t}(\sum_{j\ne t}e^{y_j})}{(\sum_{j=1}^ce^{y_j})^2}\biggr]\cdot 1
> $$
> and
> $$
>     \frac{\partial p_t}{\partial y_s} = -\frac{e^{y_t}e^{y_s}}{(\sum_{j=1}^ce^{y_j})^2} = \biggr[\frac{e^{y_t}(\sum_{j\ne t}e^{y_j})}{(\sum_{j=1}^ce^{y_j})^2}\biggr]\cdot(-\frac{e^{y_s}}{\sum_{j\ne t}e^{y_j}}) = \biggr[\frac{e^{y_t}(\sum_{j\ne t}e^{y_j})}{(\sum_{j=1}^ce^{y_j})^2}\biggr]\cdot(-\alpha_s),\quad s\ne t
> $$
>
> Thus $\phi^t(\mathbf{x})\_p = \Bigr[\frac{e^{y_t}(\sum_{j\ne t}e^{y_j})}{(\sum_{j=1}^ce^{y_j})^2}\Bigr]\cdot\big(\phi^t(\mathbf{x}) - \sum_{s\ne t}\alpha_s\phi^s(\mathbf{x})\big) =  \Bigr[\frac{e^{y_t}(\sum_{j\ne t}e^{y_j})}{(\sum_{j=1}^ce^{y_j})^2}\Bigr]\cdot\phi^t(\mathbf{x})_{\mathrm{weighted}}$.
>
> Similarly, for the Linear Approximation, we have
> $$
>     \phi^t(\mathbf{x})\_p =\sum_{s=1}^c\frac{\partial p_t}{\partial y_s}\sum_{k}\mathbf{a}^k\odot\nabla_{\mathbf{a}^k}y_s
>     =\Bigr[\frac{e^{y_t}(\sum_{j\ne t}e^{y_j})}{(\sum_{j=1}^ce^{y_j})^2}\Bigr] \sum_k\Big(\mathbf{a}^k\odot\nabla_{\mathbf{a}^k}y_t - \sum_{s\ne t}\alpha_s(\mathbf{a}^k\odot\nabla_{\mathbf{a}^k}y_s)\Big)
>     = \Bigr[\frac{e^{y_t}(\sum_{j\ne t}e^{y_j})}{(\sum_{j=1}^ce^{y_j})^2}\Bigr]\cdot \phi^t(\mathbf{x})_{\mathrm{weighted}}
> $$
> where $\mathbf{a}^k$ represents the $k$-th channel output of the CNN layers as shown in Eq (2)-(4) of our main paper.
>
> Since the proportional relation between $\phi_p$ and $\phi_{\mathrm{weighted}}$ preserves the relations among features, we abuse the ``\simeq`` symbol "$\simeq$"a little here.
>
> **Input$\times$Gradient and Integrated Gradient.**
> As mentioned in Line 206-208, Integrated Gradient has been proved to be equivalent to Input$\times$Gradient for ReLU DNNs without bias terms. This is because such NNs $F:\mathbb{R}^d\rightarrow\mathbb{R}$ are *non-negatively homogeneous* [1]. That is, $F(\alpha\mathbf{x}) = \alpha F(\mathbf{x})$ for all $\alpha\in\mathbb{R}_{\ge 0}$. Thus let $\gamma(\alpha) = \alpha\mathbf{x}$ denote the straightline path, then the IG values with zero baselines can be calculated by [1]:
> $$
>     \mathrm{IG}_i(F, \mathbf{x}) = \int_0^1\frac{\partial F(\gamma(\alpha))}{\partial \gamma_i(\alpha)}\frac{\partial\gamma_i(\alpha)}{\partial\alpha}\mathrm{d}\alpha = \int_0^1\frac{\partial F(\alpha\mathbf{x})}{\partial \alpha x_i}\frac{\partial \alpha x_i}{\partial \alpha}\mathrm{d}\alpha
>     = \int_0^1 \frac{\partial F(\alpha\mathbf{x})}{\partial \alpha x_i}x_i\mathrm{d}\alpha = \int_0^1\frac{\partial F(\mathbf{x})}{\partial x_i}x_i\mathrm{d}\alpha
>     = x_i\times\frac{\partial F(\mathbf{x})}{\partial x_i}=\mathrm{I\times G}_i(F,\mathbf{x})
> $$
>
> **Related work.**
> Thanks very much for pointing this out. By "new model structure" we mean that ProtoPNet belongs to the self-interpretable models genre. Also, the "require annotations" comes from the discussion of S9.2 in the ProtoPNet paper [2], where the number of prototypes is defined based on the annotations of CUB-200 dataset. We are sorry for the ambiguity here. We agree that ProtoPNet is not a proper related work here, and will revise this carefully in the manuscript.
>
> **References**
>
> [1] Hesse, R., Schaub-Meyer, S., \& Roth, S. (2021). Fast axiomatic attribution for neural networks. Advances in Neural Information Processing Systems, 34, 19513-19524.
>
> [2] Chen, C., Li, O., Tao, D., Barnett, A., Rudin, C., \& Su, J. K. (2019). This looks like that: deep learning for interpretable image recognition. Advances in neural information processing systems, 32.

---

> > ### Comment · Reviewer_aXtB · 2022-08-09
> > **Thank you for the response**
> >
> > Thank you for the response! Your clarification has made the equations much easier to understand. Please consider the following changes in your revision:
> >
> > 1. Instead of using the \simeq symbol, please use the \propto symbol, which is reserved for describing that something is proportional to another.
> >
> > 2. Please consider including the above derivations in the main text -- this will make it easier for readers to understand the mathematics behind your claims.
> >
> > I am keeping my positive score.

---

> > > ### Author Response · Authors · 2022-08-09
> > > **Thank You**
> > >
> > > Dear Reviewer,
> > >
> > > Thank you very much for your constructive feedback and for acknowledging our work. We will revise our final paper accordingly.
> > >
> > > Best,
> > > Authors

---

### Official Review · Reviewer_Ca7X · 2022-07-11

**Rating:** 6
**Confidence:** 3
**Soundness:** 4 excellent
**Presentation:** 3 good
**Contribution:** 2 fair

**Summary:**

The paper presents a new method for contrastive ('why class x and not class y', as opposed to 'why class x') explanations of classification networks. The main idea is to use the attributions of other classes, weighted by the softmax of their logits. The methods's applicability as an extension of existing explainability methods is shown, as well as its computational efficiency: While using information from all of the classes, ir only requires a single backward pass.

**Questions:**

* Figure descriptions - it would help if shortcuts bfa and nf from Fig. 1 were explained in the description. Also, the meaning of the colours (red/blue) is only explained in the first figure.
* Figure 4 - the names of the top 2 classes are not mentioned, which might be usefull for understandign the method. The visual comparisons only compares the proposed method to non-contrastive methods, comparison with the max/mean contrastive method could provide more insight.
* Linear approximation - not mentioned in related work, is there a reference?
* Backpropagation methods - term used in multiple places in the paper - might lead to confusion with regular back-propagation algorithms in NNs, rather than explainability
* The term 'most possible classes' used ie. in Figure 4 - are these the classes with the highest probability?
* A couple of typos/grammar mistakes: 'But without biased' on line 260, 'for examples' on line 272, 'why does the model nto classifying' on line 273, ...
* Line 303-304: Could you explain what is meant by 'It should be noticed that here all samples are covered in the experiment thus there is no uncertainty present in the results of table 2.'? Also, there is no Table 2, the reference should be to Table 1.

**Limitations:**

Limitations are properly addressed in the conclusions section.


**Strengths And Weaknesses:**

**Originality** - while the extension of existing methods is very simple, it has not been used before in the context of explainability (to the best of my knowledge). Related work is properly cited and the main contributions are well separated from previous work.

**Quality** - all the claims are well supported theoretically and the authors provide convincing experimental evaluation. The work is self-contained but proposes directions for future work.

**Clarity** - the paper is well oeganized, easy to follow and written in good English. Main ideas are explained clearly. Information about details of finetuning required to reproduce the reported numbers is missing (line 269, 'fine-tuned correspondingly').

**Significance** - the method is a simple extension of existing methods, limited to a specific task yet novel and well supported both theoretically and experimentally. The method relies on softmax weights, so is not directly applicable to ie. multilabel classification.

---

> ### Author Response · Authors · 2022-08-02
> **Response to Reviewer Ca7X**
>
> We thank the reviewer very much for the acknowledgment of our work. We answer all the questions as follows.
>
> **Information about details of finetuning required to reproduce the reported numbers is missing (line 269, "fine-tuned correspondingly").**
> In the experiments, we utilize CNN models pretrained on ImageNet-1k from ``torchvision``. By "fine-tuned correspondingly" we mean "fine-tuned corresponding to the downstream datasets (e.g. CUB-200, Flower-102, etc.)". For each downstream task, we modify the fully-connected layers so that the output dimension is consistent with the number of the dataset classes, and train the model with SGD optimizer for 200 epochs. The learning rate is initialized with 0.001 and is taken half at the 60th and the 120th epoch. The momentum is set to 0.9 and the weight decay factor is $5\times 10^{-4}$. The training losses converge for all downstream tasks. These details were skipped in the manuscript because as a modification to post-hoc explanation methods, our method does not require any specific features of the model to be explained.
>
> **Figure description.**
> Thanks! In the caption of figure 1, bfa. and nf. are abbreviations for black footed albatross and northern fulmar, respectively. As for the colors, for the heatmaps we use the ``seismic`` colormap for visualizations, hence red and blue areas indicate positive and negative values, respectively. For the text, the red and blue numbers represent the predicted probability for the corresponding classes after blurring. Red numbers represent that they **should** be larger than the black numbers before, while blue numbers represent that they **should** be smaller. We will include more detailed descriptions to the figures for better presentation.
>
> **Figure 4.**
> Thanks! Here we clarify that the two classes for the five images are 1) black footed albatross (L), sooty albatross (R); 2) frigatebird (L), laysan albatross (R); 3) sooty albatross (L), laysan albatross (R); 4) groove billed ani (L), bronzed cowbird (R); 5) parakeet auklet (L), least auklet (R). We also present more detailed visualizations to demonstrate how the weighted contrastive scheme captures contrastive features. Please see [this figure](https://drive.google.com/file/d/1GYcE63zUmGw8oFY1vjKHwbx3RIPguu1f/view?usp=sharing) for more details. We also present comparisons to the mean/max contrastive methods in this figure. For the mean contrastive method, taking the arithmetic average over $c$ classes leads to a very minor change from the standard attribution when $c$ is large (e.g. $c=200$ for CUB-200). As for the max contrastive method, it generates a similar explanation as the weighted contrastive method when the prediction is very certain (i.e. the entropy of $\mathbf{p} = \{p_1,\cdots,p_c\}$ is small), however, when there are multiple classes with relatively high predicted probabilities, the max contrastive method only captures the difference between two classes, which is not sufficient. As we argued in the main paper, when explaining "*why not other classes*", we need information from all classes instead of just the one target class. And the max contrastive method simply increases the number of target classes from 1 to 2. Please see [this figure](https://drive.google.com/file/d/1YohvIV64oUUHt5mUevaWw7FhRLwbfFNW/view?usp=sharing) for more detailed comparisons.
>
> **Linear Approximation.**
> The linear approximation is a method implemented by [``torchray``](https://facebookresearch.github.io/TorchRay/attribution.html#module-torchray.attribution.linear_approx). It calculates the element-wise product between the feature-gradient and the feature directly. As a result, it can also be viewed as "Feature$\times$Gradient" which is similar to Input$\times$Gradient. Here "feature" means the output of the CNN layers.
>
> **Nomenclature of back-propagation methods.**
> We thank the reviewer for mentioning this. The term "back-propagation methods" is to be distinctive from "perturbation methods", another main genre of post-hoc explanation methods. We follow this nomenclature since it is widely used, e.g. in [1]. We are sorry for the ambiguity and will distinguish it from the regular back-propagation algorithms.
>
> **"Most possible classes".**
> Yes, by "most possible class" and "second possible class" we mean the class with the highest probability and the second highest probability. We'll elaborate on this more in the manuscript.
>
> **Typos/Grammar issues.**
> Thanks! We will carefully revise the manuscript.
>
> **Line 303-304**
> Here "table 2" is indeed a typo and it should be "table 1". And the referred sentence simply means that the results shown in table 1 are deterministic since there's no uncertainty involved. When repeated multiple times, the results would be invariant, hence there's no need to calculate the average or standard deviation.
>
> **References**
>
> [1] Qi, Z., Khorram, S., \& Li, F. (2019, June). Visualizing Deep Networks by Optimizing with Integrated Gradients. In CVPR Workshops (Vol. 2).

---

> > ### Comment · Reviewer_Ca7X · 2022-08-08
> > **Thank you**
> >
> > Thank you for the clarifications and addressing all the issues - I am keeping my positive score.

---

> > > ### Author Response · Authors · 2022-08-09
> > > **Thanks**
> > >
> > > Thanks very much for the acknowledgment of our work!

---

### Author Response · Authors · 2022-08-09
**We are happy to answer more questions if there still exist concerns for our paper.**

Dear Reviewers,

Thanks for your time and efforts in reviewing our paper. We appreciate your constructive comments. Hopefully, our response can address your concerns.

If you have further questions or confusion, we would be very happy to clarify. Thank you very much.

Best,
Authors

---

### Meta-Review · Area_Chair_4Ud6 · 2022-08-27

**Recommendation:** Accept
**Confidence:** Certain

**Metareview:**

Reviewers expressed overwhelmingly positive opinions about the simple, easily implementable, and at the same time innovative procedure proposed in the paper for obtaining gradient-based class-contrastive explanations. Appreciation also transpired for the significance of this work in clarifying some technical points in the gradient-based XAI literature and the potential for future work that the paper opens up.
One of the main criticisms raised in the reviews, the lack of comparisons against other contrastive explanation methods, has been addressed satisfactorily with additional experiments and discussions in the rebuttals.
The most important remaining criticism was a doubt on the merits of one of the key technical points in the paper regarding whether gradient-based attributions should be computed according to the softmax outputs or logits. Reviewers pointed out that computing attributions with respect to softmax outputs instead of logits is already common practice in the field.
Reviewers expressed strongly the opinion that it would be appropriate to characterizing and clarify this situation, as it could be potentially misleading and indeed counterproductive even for the paper to denote attribution methods with respect to logits as "standard", while it's instead the case that some implementation of gradient-based attribution methods already attribute with respect to the softmax output (albeit inconsistently).
In conclusion, the reviewing panel voted for accepting the paper, under the condition that the camera-ready version of the paper explicitly clarify the distinction between the two approaches and discuss the implication of choosing one of the other, without however referring to attribution with respect to logits as standard, but merely pointing out that until now the distinction has been vague and implementations inconsistent. From this technical standpoint, Reviewers ask that the contribution of the paper should then be explicitly characterized as clarifying the distinction between logits and softmax attributions, rather than as the proposal of a new procedure in opposition to an already established standard. This is already perceived as a strong contribution to the community, as phrasing it specifically as indicated would help elucidate the state of affairs in the literature and make the community aware of this outstanding blind-spot.

**Award:**

No

---

### Decision · Program_Chairs · 2022-09-14

Accept